# Rational Screening of High-Voltage Electrolytes and Additives for Use in LiNi_0.5_Mn_1.5_O_4_-Based Li-Ion Batteries

**DOI:** 10.3390/molecules27113596

**Published:** 2022-06-03

**Authors:** Oleg A. Drozhzhin, Vitalii A. Shevchenko, Zoia V. Bobyleva, Anastasia M. Alekseeva, Evgeny V. Antipov

**Affiliations:** 1Department of Chemistry, Lomonosov Moscow State University, 119991 Moscow, Russia; vitalii.shevchenko@skoltech.ru (V.A.S.); alekseevaam@gmail.com (A.M.A.); evgeny.antipov@gmail.com (E.V.A.); 2Skoltech Center for Energy Science and Technology, Skolkovo Institute of Science and Technology, Nobel Str. 3, 143026 Moscow, Russia; 3Department of Material Science, Lomonosov Moscow State University, 119991 Moscow, Russia; zoyamostovik@gmail.com

**Keywords:** Li-ion batteries, high-voltage electrolyte, electrolyte additives, concentrated electrolytes, LiNi_0.5_Mn_1.5_O_4_

## Abstract

In the present work, we focus onthe experimental screening of selected electrolytes, which have been reported earlier in different works, as a good choice for high-voltage Li-ion batteries. Twenty-four solutions were studied by means of their high-voltage stability in lithium half-cells with idle electrode (C+PVDF) and the LiNi_0.5_Mn_1.5_O_4_-based composite as a positive electrode. Some of the solutions were based on the standard 1 M LiPF_6_ in EC:DMC:DEC = 1:1:1 with/without additives, such as fluoroethylene carbonate, lithium bis(oxalate) borate and lithium difluoro(oxalate)borate. More concentrated solutions of LiPF_6_ in EC:DMC:DEC = 1:1:1 were also studied. In addition, the solutions of LiBF_4_ and LiPF_6_ in various solvents, such as sulfolane, adiponitrile and tris(trimethylsilyl) phosphate, atdifferent concentrations were investigated. A complex study, including cyclic voltammetry, galvanostatic cycling, impedance spectroscopy and ex situ PXRD and EDX, was applied for the first time to such a wide range of electrolytesto provide an objective assessment of the stability of the systems under study. We observed a better anodic stability, including a slower capacity fading during the cycling and lower charge transfer resistance, for the concentrated electrolytes and sulfolane-based solutions. Among the studied electrolytes, the concentrated LiPF_6_ in EC:DEC:DMC = 1:1:1 performed the best, since it provided both low SEI resistance and stability of the LiNi_0.5_Mn_1.5_O_4_ cathode material.

## 1. Introduction

Works aimed at increasing the average potential of lithium-ion batteries (LIBs) have been ongoing over the past few decades. The fundamental possibility for the potential increasing is based on the existence of a number of cathode materials with an average lithium extraction/insertion potential of more than 4.5 V vs. Li/Li^+^, such as LiNi_0.5_Mn_1.5_O_4_, LiCoPO_4_ and Li_2_CoPO_4_F [1,2,3,4]. The former is one of the most studied materials and is close to being industrially used as cathode material [5]. LiNi_0.5_Mn_1.5_O_4_ is characterized by a theoretical capacity of 147 mAh g^−1^ due to reversible Ni^2+^/Ni^4+^ redox, a two-stage voltage-composition plateau, indicating the occurrence of two two-phase transitions at an average potential of 4.75 V, and high rates of 3D Li^+^ diffusion within the structure [6,7,8,9,10,11]. However, the use of a standard Li-ion electrolyte inevitably leads to the rapid degradation of LiNi_0.5_Mn_1.5_O_4_-based half- and full-cells. The development of the new improved electrolyte formulations can overcome this problem. 

In general, there are three main directions in the development of liquid electrolytes that are stable in the high-voltage region: (1) adding a small amount (typically ~1%) of additives to the standard electrolyte solution, (2) the use of stable solvents combined with corresponding salts (including works on ionic liquids) and (3) the variation of the salt concentration. In the first direction, the most well-studied additives are fluoroethylene carbonate (FEC), lithium bis(oxalate) borate (LiBOB), lithium difluoro(oxalate)borate (LiDFOB), vynylene carbonate (VC), tris(trimethylsilyl) borate (TMSB), tris(trimethylsilyl) phosphite (TMSP), prop-1-ene-1,3-sultone (PES) and many others [12,13,14,15,16,17,18,19,20,21,22,23]. It is worth noting that some compounds (such as LiBOB, LiDFOB and FEC, adiponitrile) may be used both as the additives and the basic component of the salt or solvent. It is believed that the main contribution of additives to the improvement of the high-voltage stability of electrolytes is the formation of a stable interface at the cathode. The works conducted in the second direction were aimed at improving the intrinsic stability of the solvent and solvents such as sulfones, nitriles and fluorinated alkyl carbonates were chosen as objects of study [24,25,26,27,28,29,30,31,32,33]. Moreover, researchers studied high-concentrated systems (the third direction) and observed an increase in the ionic association degree and decrease in the amount of free solvent molecules as the major factors to improve the electrochemical properties of electrolytes. Since 2016, several papers devoted to the high-voltage oxidative stability of the concentrated electrolytes were published [34,35,36,37,38,39,40]. The usual lithium salts used in high-concentrated electrolyte research contain large anions providing a high dissociation degree, such as bis(fluorosulfonyl)amide (FSA-), bis(fluorosulfonyl)imide (FSI-) and bis(trifluoromethanesulfonyl)imide (TFSI-) [34,36,40]. However, these salts are rather expensive and may cause severe problems to the Al current collector. “Traditional” lithium salts, such as LiPF_6_ and LiBF_4_, are also used in spite of their lower solubility limit, and also provide an increase in the oxidative stability of the electrolytes [35,37,38,39]. Inaddition to better oxidation resistance, the concentrated electrolytes are known to improve the kinetics of the charge transfer between electrode and electrolyte, as well as the ionic conductivity of the latter [40,41,42]. Thus, a Li^+^ hopping conduction mechanism characterized by high Li^+^ self-diffusion coefficients has been recently reported for the concentrated LiBF_4_/sulfolane system [43,44]. Unfortunately, data on the high-voltage stability of this system have not been presented in the literature to date. 

Despite the impressive number of studies on high-voltage electrolytes, the analysis of the competitive advantages of one or another development approach, and the choice of the high priority ones are hampered by the fact that each research work studied a selected set of materials/methods/protocols, etc. In the present work, we compare the high-voltage electrochemical performance of electrolytes prepared in accordance with previously published approaches within the three major directions mentioned above. Therefore, in this paper, we study the influence of several popular and easily accessible additives on the stability of the standard Li-ion electrolyte (1 M LiPF_6_ in EC:DMC:DEC = 1:1:1) and the stability of the electrolytes based on different solvents and concentrated solutions. The data obtained help to formulate a promising approach in the development of sustainable, affordable and efficient electrolytes for next-generation lithium-ion batteries.

## 2. Results

The preliminary analysis of the anodic stability of the prepared electrolytes at high potentials was conducted by means of CV with idle electrodes (composed of carbon black and PVDF). The results are shown in Appendix A and Figure 1 and Table 1.

The main findings canbe summarized as follows:(a)Increasing the concentration of LiPF_6_ in the standard solvent composition (EC:DEC:DMC = 1:1:1) leads to a decrease inthe anodic current at elevated (>5.0 V vs. Li/Li^+^) potentials. However, it also leads to a severe increase in the anodic current at moderate potentials (above 3.5 V vs. Li/Li^+^) up to several µA. We believe that this response is a consequence of the formation of surface layers, which, most likely, further protect the electrolyte from decomposition at high potentials.(b)The use of 1% FEC, PES and ADN improves the stability up to 5.3 V vs. Li/Li^+^. The additives of VC and LiBOB reduce the oxidative current at 4.5 V and, therefore, may be useful for any cathode materials with a corresponding voltage limit; however, their use in the high-voltage systems is questionable. LiDFOB in an amount of 1% causes a strong oxidation of the electrolyte by 5.3 V vs. Li/Li^+^. A decrease in the concentration of LiDFOB by a factor of 20 leads to a decrease in the anode current, but the current value remains significant.(c)The most interesting results were demonstrated by the solutions of LiBF_4_ in SL with a concentration of 1 M and more concentrated ones. The use of ADN as a solvent also significantly improved the high-voltage stability of the electrolyte.

Based on the results presented above and the literature data, we chose nine electrolyte solutions for further investigation with the high-voltage cathode material LiNi_0.5_Mn_1.5_O_4_. The CVs of these nine solutions are combined in Figure 1. 

The LiNi_0.5_Mn_1.5_O_4_ cathode material was obtained via the co-precipitation method, followed by the hydrothermal treatment of the Ni_0.25_Mn_0.75_CO_3_ intermediate and high-temperature annealing with the Li source. PXRD revealed the presence of the single cubic spinel phase (sp.gr. *Fd*$\bar{3}$*m*, *Z* = 8, *a* = 8.1660(1) Å, *V* = 544.5(1) Å^3^). The sample consisted of spherical aggregates (2~5 µm in size) formed by small crystallites (200~500 nm in size), as it can be observed in the SEM images (Figure 2b). The Mn/Ni ratio determined by EDX amounted to 3.2(1), which corresponds to the LiNi_0.48(2)_Mn_1.52(2)_O_4_ composition. The sample demonstrates a good electrochemical performance with a reversible capacity of ~140 mAh g^−1^ depending on the electrolyte composition. An example of a charge–discharge curve (1 M LiBF_4_ in SL electrolyte, C/10 rate) is presentedin Figure 2c.

The galvanostatic cycling of the LiNi_0.5_Mn_1.5_O_4_ cathode material in the selected electrolyte solutions was performed in three successive stages: 10 cycles at C/10 rate, 30 cycles at C/3 rate and 100 cycles at 1C rate (Figure 3). After each stage, EIS spectra were collected, as well as for the as-prepared cells (Figure 4). All EIS experiments were performed at the discharged state of the working electrode. 

The following issues were revealed:(a)The LiNi_0.5_Mn_1.5_O_4_-based working electrodes demonstrate degradation in all the studied electrolyte solutions. However, in some cases, a clearly improved anodic stability is observed. A group of leaders at low (C/10 and C/3) cycling rates includes 3 M LiPF_6_ in EC:DEC:DMC = 1:1:1 (Standard 3 M), 1 M and 3 M LiBF_4_ in SL, 1 M LiBF_4_ in AND:EC = 1:1,standard + PES and—unexpectedly—standard + LiDFOB electrolytes. The latter, taken in trace amounts, was included in the studied group based on the literature data in spite of the high oxidation current at CV. (b)The Coulombic efficiency of all the studied cells falls short of the needs for real application. The best value at the low charge–discharge rate of C/10 (92%) was demonstrated by Standard 3 M electrolyte; at 1C, the current density efficiency of the cell increased up to 98%. It should be noted that the Coulombic efficiency of all the cells varied from cycle to cycle, probably due to the growth of surface layers from the degradation products. In most cases, it leads to two mutually directed (positive and negative) trends: on the one hand, the degree of electrolyte degradation decreases because of surface passivation, and on the other hand, the charge transfer resistance at the interface increases, which leads to faster a degradation of the capacity. The latter trend was studied in detail by means of EIS.


EIS spectra were collected at four different stages of the cell cycling: as-assembled, after 10 cycles at C/10 rate, after 30 cycles at C/3 rate and after 100 cycles at 1C rate. All experiments were performed at 100% depth-of-discharge (E = 2.8 V vs. Li/Li^+^). The Nyquist plots for the first and the last stages are presented at Figure 4a,b. We focused on the analysis of the charge transfer resistance, which is expressed in Nyquist plots as a characteristic semicircle, having an equivalent in the form of a parallel resistance and capacitance (or “constant phase element”, CPE) and being previously described and widely used for the characterization of the electrode–electrolyte interface [45]. EIS experiments divided the studied electrolytes into two groups. The first group includes solutions that are characterized by an increase in charge transfer resistance (R_ct_) throughout the cycling: Standard, Standard + FEC, Standard + LiDFOB. In these cases, the resistance increased by 5–7 times, from 100–200 Ohms to 700–1400 Ohms, with the largest increase occurring in the first ten charge–discharge cycles. Obviously, an increase in the resistance is associated with a steady increase in the amount of electrolyte oxidation products on the cathode surface during cycling. Other electrolytes demonstrate an interesting feature of decreasing R_ct_ values after cycling at elevated C-rates (C/3 or 1C). For example, the cell with 1 M LiBF_4_ in SL demonstrated growth in the resistance from 20 to 450 Ohms during the first ten cycles, but after 30 cycles at C/3 rate and 100 cycles at 1C rate, this value decreased to 210 and 160 Ohms, correspondingly. This could be a consequence of the slightly different mechanism of SEI formation at different current densities due to the occurrence of competing processes with distinct rate constants. A large number of different species were found on the cathode surface in previous studies, including lithium carbonate, semicarbonates, fluoride, organophosphates and polymers [46]. The ratio between these components determines the electrochemical properties and the stability of the cathodic SEI. It is obvious that the variation in the cycling conditions of the cathode material can change the amount of one or another component in the SEI, since the mechanisms and rates of their formation differ for each type of reaction products. Moreover, as shown earlier, in contrast to the SEI on the anode, the cathode interface is not so stable and tends to dissolution/formation on each charge–discharge cycle, at least at moderate oxidative potentials [46].

The observed trends in charge transfer resistance behavior match perfectly with the capacity retention at cycling. The 3 M LiPF_6_ in EC:DEC:DMC = 1:1:1 (Standard 3 M) electrolyte provides the lowest R_ct_ and the best cyclability of the LiNi_0.5_Mn_1.5_O_4_ half-cell. The top three is closed by 1 M and 3 M LiBF_4_ in SL. It is worth noting the positive effect of PES on the high-voltage stability of a standard electrolyte: it is the only additive that enabled the decrease in R_ct_ after cycling at the 1C current density.

To analyze the chemical stability of the cathode material in the studied electrolytes, we performed ex situ PXRD and EDX investigations of the electrodes after cycling. To the best of our knowledge, such characterization of materials is extremely rare in works devoted to electrolytes, so we considered this study to be of utmost importance. Surprisingly, there are no noticeable signs of material degradation on the PXRD patterns (Figure 5a). The values of the unit cell volume of LiNi_0.5_Mn_1.5_O_4_ combined with Mn/Ni determined by EDX are presented in Figure 5b. 

The examination of the electrodes’ surface after cycling with SEM (Figure 5c) also revealed no visible signs of degradation. We can only note a more noticeable contribution of surface films in the case of concentrated electrolytes and a partial pulverization of surface particles in the PES-containing electrolyte. However, it cannot be ruled out that the electrolyte decomposition products were dissolved during the washing of the electrodes after their removal from the cells. 

As it can be seen in Figure 5b, the values of the unit cell volume of the LiNi_0.5_Mn_1.5_O_4_ cathode material cycled in the electrolytes based on the standard (LiPF_6_ in EC:DEC:DMC = 1:1:1) solution are close to those of the initial material. However, the cycling in the LiBF_4_-based solutions leads to some increase in the unit cell volume, although this change is not significant. The maximum (~0.6% volume change) is achieved for the most concentrated electrolyte, 3 M LiBF_4_ in SL. It is worth noting that this electrolyte is also characterized by the strongest deviation of the Mn/Ni ratio from the initial value. We may assume that the use of LiBF_4_ as an electrolyte salt may lead to partial Mn dissolution, although, electrochemically, these solutions, especially 1 M and 3 M LiBF_4_ in SL, demonstrate excellent stability.

## 3. Discussion

Of interest is the fact that this suggestion differs from the previously published data, in which the main reason for the dissolution of manganese cations was assumed to be the decomposition of DEC, accompanied by release of HF [47]. It is also interesting that, according to our data, the spinel LiNi_0.5_Mn_1.5_O_4_ cathode material itself is quite stable, even in those electrolytes where the capacity drop during the cycling turned out to be the strongest, for example, in the standard or “Standard+FEC” solutions.

To summarize the results obtained, we can conclude that the concentrated electrolyte solutions or solutions stable towards oxidation, such as sulfolane, exhibit a better electrochemical stability than the electrolytes with different types of additives. Lithium cells with the LiNi_0.5_Mn_1.5_O_4_ cathode material demonstrate the best cycling stability in 3 M LiPF_6_ in EC:DEC:DMC = 1:1:1 (Standard 3 M), 1 M and 3 M LiBF_4_ in SL electrolytes. These and some other solutions exhibit an interesting feature of a decrease in the charge transfer resistance after cycling at elevated current rates. From the other side, the ex situ material characterization revealed that the degradation of the cell with electrolytes based on the LiPF_6_ in EC:DEC:DMC = 1:1:1 solution is not associated with the degradation of the active material, but solely with an increase in the resistance of the surface layers at the cathode–electrolyte interface. Therefore, the concentrated solution of LiPF_6_ in EC:DEC:DMC = 1:1:1 provides both low SEI resistance and stability of the LiNi_0.5_Mn_1.5_O_4_ material and can be considered as the optimal choice for the development of a new generation of high-voltage LIBs. The C-rate retention tests conducted for the LiNi_0.5_Mn_1.5_O_4_ cathode in half-cells with 3 M LiPF_6_ in EC:DEC:DMC = 1:1:1 (Standard 3 M) electrolyte (Figure 6) demonstrate the possibility of both fast charge and fast discharge up to 5C rates without significant capacity fading (in both cases, at least 120 mAh·g^−1^ may be gained at 5C current rate). However, it is worth noting that the most preferable charge rate is C/3~1C since it does not require an increase in the anode potential limit above 4.9 V. The relatively low resistance of the electrolyte and the cathode–electrolyte interface, which makes it possible to achieve stable cycling at high current densities, distinguishes liquid electrolytes from ceramic and polymer analogues, which are currently considered as alternative technologies. However, in most works, the properties of these systems are studied using low currents due to kiloohmic cell resistances, and an increase in efficiency is possible only at elevated temperatures [48]. In addition, a number of articles reported on the poor stability of solid electrolytes in the high-voltage potential region due to the growth of highly resistive interfacial layers [49,50]. Thus, it can be concluded that liquid electrolytes are the most preferred candidates for use in high-voltage lithium-ion systems for the foreseeable future.

## 4. Materials and Methods

### 4.1. Synthesis

The LiNi_0.5_Mn_1.5_O_4_ cathode material was prepared using solvothermal synthesis under conditions similar to those previously described [51]. NiSO_4_·6H_2_O (>98%) and MnSO_4_·H_2_O (>98%) were dissolved in water in a molar ratio of 1:3. A sodium carbonate (>99.5%) solution was added; the obtained suspension was heated to 140 °C and conditioned at this temperature for 12 h. The carbonate precipitate was washed, dried and mixed with Li_2_CO_3_ (>99%) taken with 5% excess. The precursor was annealed at 350 °C for 2 h and at 800 °C for 10 h with intermediate grinding.

### 4.2. Material Characterization

A Panalytical Aeris Research diffractometer (CuKα radiation, Bragg–Brentano geometry, PiXCel detector, total angular range of 3–1202θ, a step size of ca. 0.005° and variable exposure time, Almelo, The Netherlands) was used for powder X-ray diffraction (PXRD) measurements. For the ex situ PXRD investigation, the electrodes were preliminary washed with propylene carbonate (PC) under argon, dried and covered by X-ray amorphous Kapton tape. PXRD data refinements were performed using the JANA2006 program package [52]. The particle size, morphology and cationic composition were investigated by means of a JEOL JSM-6490LV scanning electron microscope (Tokyo, Japan) equipped with EDX spectrometer INCA X-Sight (Oxford Instruments, Oxford, UK).

### 4.3. Electrochemical Studies

All the electrolyte components were purchased from Merck (formerly Sigma Aldrich, Rahway, NJ, USA) with the highest purity available and, additionally, they were preliminary dried before use. Li-ion electrolyte solutions were prepared by dissolving the appropriate amount of the corresponding salt (LiPF_6_ or LiBF_4_) in the solvent (EC:DMC:DEC = 1:1:1 or SL (sulfolane) or ADN (adiponitrile)). In some cases, an appropriate amount of the additives (0.3 or 1 wt.%), including FEC, LiBOB, LiDFOB, VC and PES, was used. LiPF_6_ or LiBF_4_ were kept under dynamic vacuum at 60 °C for 24 h. The solvents were dried with activated 4 Å molecular sieves. The following compositions were prepared:

1 M LiPF_6_ in EC:DEC:DMC = 1:1:1

2 M LiPF_6_ in EC:DEC:DMC = 1:1:1

3 M LiPF_6_ in EC:DEC:DMC = 1:1:1

1 M LiPF_6_ in EC:DEC:DMC = 1:1:1 + 1% FEC

1 M LiPF_6_ in EC:DEC:DMC = 1:1:1 + 1% VC

1 M LiPF_6_ in EC:DEC:DMC = 1:1:1 + 1% LiBOB

1 M LiPF_6_ in EC:DEC:DMC = 1:1:1+ 1% LiDFOB

1 M LiPF_6_ in EC:DEC:DMC = 1:1:1 + 0.05% LiDFOB

1 M LiPF_6_ in EC:DEC:DMC = 1:1:1 + 1% PES

1 M LiPF_6_ in EC:DEC:DMC = 1:1:1 + 1% ES

1 M LiPF_6_ in EC:DEC:DMC = 1:1:1 + 1% ADN

1 M LiPF_6_ in EC:DEC:DMC = 1:1:1 + 1% SCN

1 M LiPF_6_ in EC:DEC:DMC = 1:1:1 + 1% TMP

1 M LiPF_6_ in EC:DEC:DMC = 1:1:1 +1% TMPi

1 M LiBF_4_ in SL

2 M LiBF_4_ in SL

3 M LiBF_4_ in SL

4 M LiBF_4_ in SL

5 M LiBF_4_ in SL

1 M LiBF_4_ in ADN

1 M LiBF_4_ in EC:DEC

1 M LiBF_4_ in TMP 

1 M LiBF_4_ in DEC:FEC = 1:1

0.6 M LiBF_4_: 0.6 M LiDFOB in EC:DEC:DMC = 1:1:1 

Where EC is ethylene carbonate (>99%), DEC is diethyl carbonate (>99%), DMC is dimethyl carbonate (>99%), FEC is fluoroethylene carbonate (>99%), VC is vinylene carbonate (>99.5%), PES is 1,3-propanesultone (>99%), ES is ethylene sulfite (>98%), ADN is adiponitrile (>99%), SCN is succinonitrile (>99%), TMP is tris(trimethylsilyl) phosphate (>99%), TMPi is tris(trimethylsilyl)phosphite (>95%), SL is sulfolane (>99%), LiPF_6_ is lithium hexafluorophosphate (>99,99%), LiBF_4_ is lithium tetrafluoroborate (>99.99%), LiBOB is lithium bisoxalatoborate (n/d) and LiDFOB is lithium difluoroxalatoborate (n/d). The solvent–salt ratio designated as «nM» was calculated as the ratio of the molar amount of salt and the volume of solvent.

The LiNi_0.5_Mn_1.5_O_4_-based electrodes were prepared by mixing 85 mass.% of the active compound, 7.5% of carbon black (Timcal Super C 65) and 7.5% of polyvinylidene fluoride (PVDF, Solvay Solef 5130) binder in N-methylpyrrolidone and spreading it on an aluminum foil by the doctor blade technique. The mass loading of the active material was appr. 6 mg/cm^2^. Idle electrodes contained 70 mass.% of carbon black and 30 mass.% of PVDF without any other active material. The dried electrodes were rolled, punched to round discs and dried at 110 °C for 3 h under dynamic vacuum. Two-electrode coin-type cells were assembled in argon-filled glove box (MBraun). Lithium metal was used as the counter electrode. Cyclic voltammetry (CV, 2.5–5.3 V vs. Li/Li^+^, 0.05 mV s^−1^), impedance spectroscopy (EIS, 100 kHz—0.05 Hz, 10 mV amplitude) and galvanostatic cycling (GC, 2.8–4.9 V vs. Li/Li^+^, C/10-1C rate) were performed using Elins P-20X8 and Elins P45X potentiostat-galvanostats (ES8 software, Chernogolovka, Russia).

## Figures and Tables

**Figure 1 molecules-27-03596-f001:**
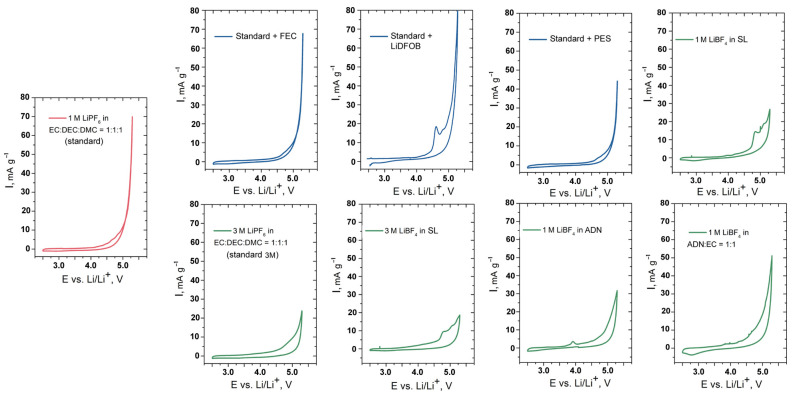
CV curves of the selected electrolytes with “idle” electrodes. The mass fraction of all additives was 1% with the exception of LiDFOB (0.05%).

**Figure 2 molecules-27-03596-f002:**
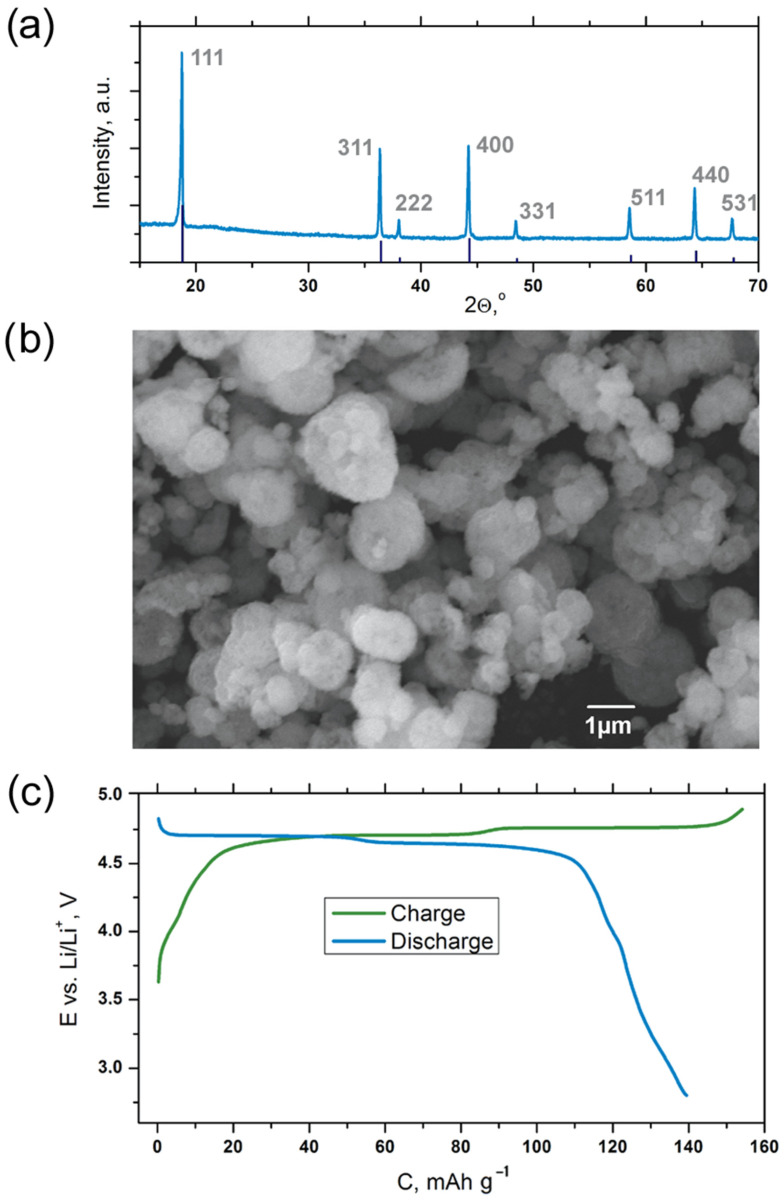
PXRD pattern (**a**), SEM image (**b**) and galvanostatic charge–discharge curve (**c**) obtained for the LiNi_0.5_Mn_1.5_O_4_ cathode material.

**Figure 3 molecules-27-03596-f003:**
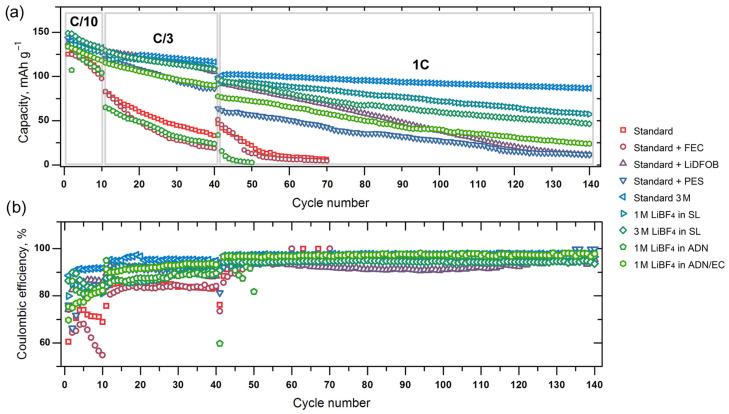
Dependence of the discharge capacity (**a**) and Coulombic efficiency (**b**) of LiNi_0.5_Mn_1.5_O_4_-based lithium half-cells cycled at C/10, C/3 and 1C rates on the number of cycle.

**Figure 4 molecules-27-03596-f004:**
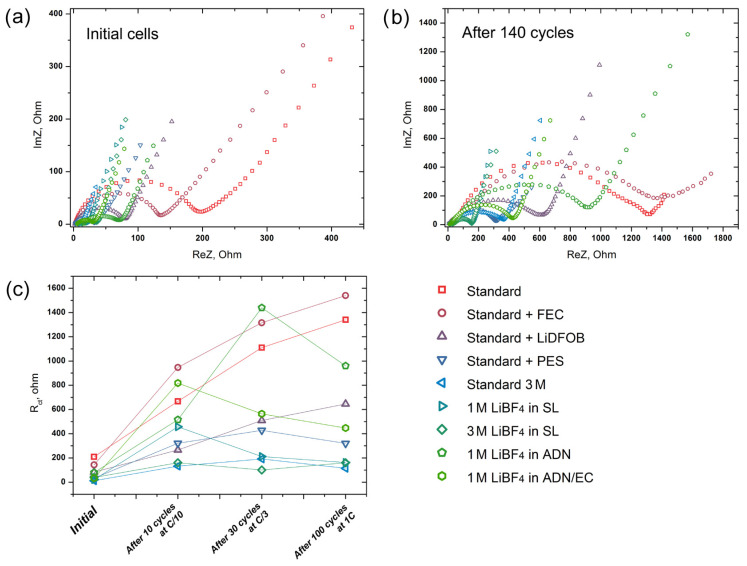
EIS Nyquist plots for the as-assembled cells (**a**) and after complete galvanostatic cycling: 10 cycles at C/10 rate, 30 cycles at C/3 rate and 100 cycles at 1C rate (**b**). Values of the charge transfer resistance (R_ct_) at different stages of the cell cycling (**c**).

**Figure 5 molecules-27-03596-f005:**
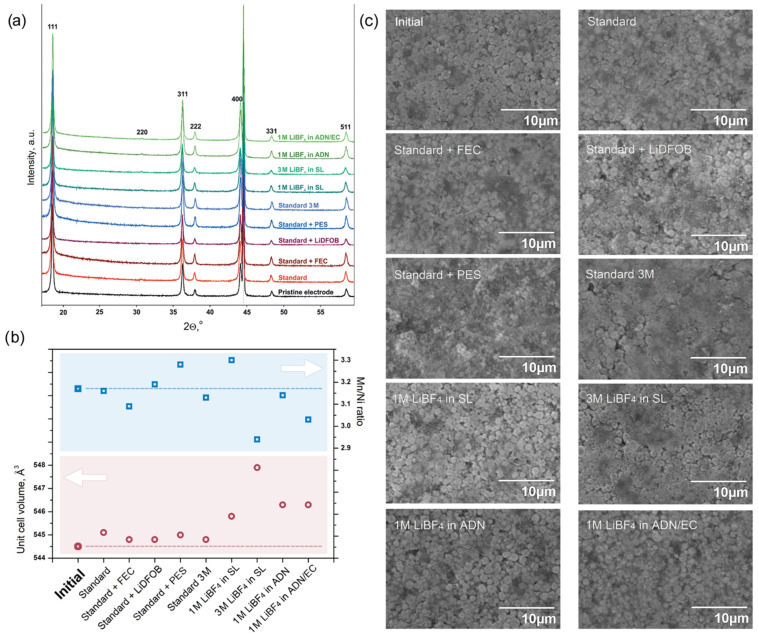
(**a**) Selected region of the ex situ PXRD patterns obtained for the pristine LiNi_0.5_Mn_1.5_O_4_ cathode and for cathodes after 140 charge–discharge cycles in the selected electrolytes. The dotted line marks the Al current collector diffraction maximum. The reflections of cubic spinel LiNi_0.5_Mn_1.5_O_4_ are indexed. (**b**) Unit cell volume and the Mn/Ni ratio of the LiNi_0.5_Mn_1.5_O_4_ cathode material after 140 charge–discharge cycles in the selected electrolytes. (**c**) SEM images of the electrodes’ surface after cycling.

**Figure 6 molecules-27-03596-f006:**
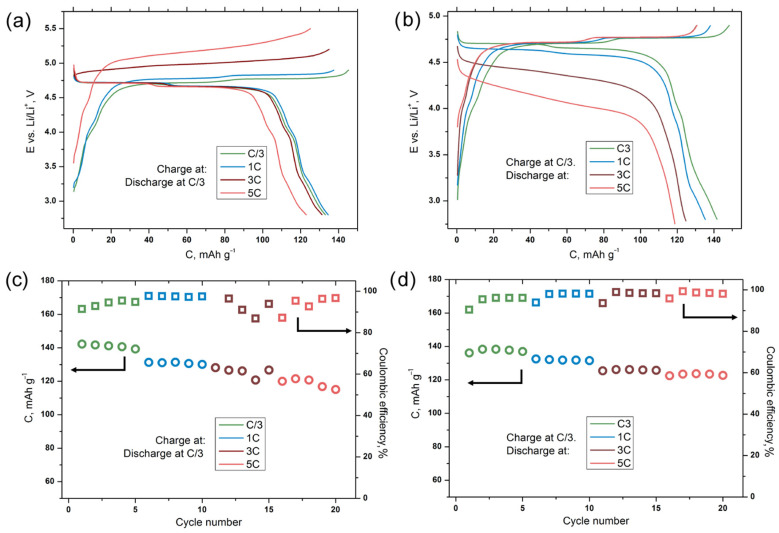
Results of the C-rate retention tests for the LiNi_0.5_Mn_1.5_O_4_ cathode in half-cells with 3 M LiPF_6_ in EC:DEC:DMC = 1:1:1 (Standard 3 M) electrolyte. (**a**,**c**) “Fast charge–slow discharge” regime; (**b**,**d**) “slow charge–fast discharge” regime.

**Table 1 molecules-27-03596-t001:** Specific anodic current at three points (4.5, 5.0 and 5.3 V vs. Li/Li^+^) during the first cycle of CV measured for the electrolytes with “idle” (C + PVDF) working electrodes.

Electrolyte Composition	Specific Anodic Current Normalized on the Mass of Carbon Black (mA g^−1^) at:		
	4.5 V vs. Li/Li^+^	5.0 V vs. Li/Li^+^	5.3 V vs. Li/Li^+^
1 M LiPF_6_ in EC:DEC:DMC = 1:1:1	2.9	11.9	69.8
2 M LiPF_6_ in EC:DEC:DMC = 1:1:1	3.1	9.1	39.1
3 M LiPF_6_ in EC:DEC:DMC = 1:1:1	4.5	10.9	38.2
1 M LiPF_6_ in EC:DEC:DMC = 1:1:1 + 1% FEC	4.3	13.7	49.4
1 M LiPF_6_ in EC:DEC:DMC = 1:1:1 + 1% VC	2.4	262.2	350.4
1 M LiPF_6_ in EC:DEC:DMC = 1:1:1 + 1% LiBOB	2.3	14.9	81.0
1 M LiPF_6_ in EC:DEC:DMC = 1:1:1 + 1% LiDFOB	3.1	21.3	167.2
1 M LiPF_6_ in EC:DEC:DMC = 1:1:1 + 0.05% LiDFOB	4.8	21.8	91.1
1 M LiPF_6_ in EC:DEC:DMC = 1:1:1 + 1% PES	2.7	12.0	41.5
1 M LiPF_6_ in EC:DEC:DMC = 1:1:1 + 1% ES	3.5	14.2	73.7
1 M LiPF_6_ in EC:DEC:DMC = 1:1:1 + 1% ADN	2.3	12.4	45.9
1 M LiPF_6_ in EC:DEC:DMC = 1:1:1 + 1% SCN	4.0	14.3	107.6
1 M LiPF_6_ in EC:DEC:DMC = 1:1:1 + 1% TMP	4.8	16.2	80.8
1 M LiPF_6_ in EC:DEC:DMC = 1:1:1 +1% TMPi	2.9	20.1	114.1
1 M LiBF_4_ in SL	3.4	17.3	26.8
2 M LiBF_4_ in SL	4.7	14.3	24.4
3 M LiBF_4_ in SL	3.9	11.5	17.9
4 M LiBF_4_ in SL	2.6	8.4	15.0
5 M LiBF_4_ in SL	3.1	9.3	18.5
1 M LiBF_4_ in ADN	3.8	13.3	31.8
1 M LiBF_4_ in EC:ADN	5.6	18.9	48.8
1 M LiBF_4_ in TMP	4.0	21.5	47.2
1 M LiBF_4_ in DEC:FEC	5.0	16.0	26.8
0.6 M LiBF4: 0.6 M LiDFOB in EC:DEC:DMC = 1:1:1	6.5	41.3	606.1

## Data Availability

Not applicable.

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
