# Peer review of "Rational Screening of High-Voltage Electrolytes and Additives for Use in LiNi0.5Mn1.5O4-Based Li-Ion Batteries"

_molecules, 2022, doi:10.3390/molecules27113596_

Round 1

Reviewer 1 Report

Manuscript ID: molecules-1700336

Title: Rational screening of the high-voltage electrolytes and additives for use in LiNi0.5Mn1.5O4-based Li-ion batteries

Reviewer Comments: (Minor Revision)

In this work, the authors prepared screeing the best electrolyte for LiNi0.5Mn1.5O4 based Li-ion batteries. After careful evaluation of the manuscript, I find that this paper would be suitable for publication in molecules, and minor revision is needed to be considered.

  1. As we know, there have been lots of electrolyte and additive are used for Li-ion batteries. From this point of view, this work can comparing various electrolyte for LiNi5Mn1.5O4 electrode. Authors should provide novelty of this manuscript.
  2. Authors can provided the SEM image of electrode for best electrolyte after the stability test and comment on the morphological degradation.
  3. If possible, authors can make a full pouch cell for the best electrolyte, evaluate the electrochemical performance and compared with commercial LiNi0.5Mn1.5O4 based Li-batteries. 

Author Response

We thank the Reviewer for the analysis of our manuscript

  1. As we know, there have been lots of electrolyte and additive are used for Li-ion batteries. From this point of view, this work can comparing various electrolyte for LiNi5Mn1.5O4 electrode. Authors should provide novelty of this manuscript.

We’ve added additional information into the Introduction to validate the novelty of the work.

  1. Authors can provided the SEM image of electrode for best electrolyte after the stability test and comment on the morphological degradation.

We’ve added these data.

  1. If possible, authors can make a full pouch cell for the best electrolyte, evaluate the electrochemical performance and compared with commercial LiNi0.5Mn1.5O4 based Li-batteries.

We are working on optimization of the electrodes preparation process, the types and ratio of the binder/conductive additives, thickness of the layers and the balance of the cathode and anode capacities. However, this work requires a long time and is, mainly, technological. We’ll try to include such data in our further publication but this manuscript is primarily directed to the laboratory investigation of the high voltage electrolyte performance.

Reviewer 2 Report

The article written by Drozhzhin et al. presents the work on the utilization of LNMO high-voltage cathode material in Li-ion batteries and studies with various common-used electrolytes and their additives. The article is very interesting and to be published in a high-impact factor journal like Molecules MDPI needs more work. I recommend a major revision of the manuscript after my suggestions and recommendations.

  1. In the whole article the language should be checked.
  2. The statement - novelty of the work should already be underlined to be more attractive for the readers: in the abstract, the whole manuscript, and the conclusions.
  3. All used chemicals to the synthesis should be described more precisely: purity, producer etc.
  4. All equipment used for the measurements should be described more precisely: name, producer, country of the producer.
  5. All presented figures are of very low quality.
  6. Introduction: in this section is not well-discussed about the utilization of different forms of high-voltage materials used as electrode in Li-ion batteries. I suggest to rewrite it, and compare the results with other studies, such as: B.-Y. Lee et al. Ceram. Int. 46 (2020) 20856-20864, G. Liang et al. J. Mater. Chem. A 8 (2020) 15373-15398.
  7. Please describe more precisely the EIS results.
  8. Provide active material loading of LiNi0.5Mn1.5O4 (in mg/cm2) in the electrode prepared.
  9. What was the conditions used for the EIS tests, only room temperature? If yes, I suggest to perform the whole EIS analysis in high- and low-temperature conditions and re-analyze the electrolyte and additives to select the best one which could be a potential for Li-ion batteries applications.
  10. High-rate tests (at 5 C, 10 C, 20 C etc.) are crucial to complete the idea of this work.

Author Response

We thank the Reviewer for the analysis of our manuscript.

1. In the whole article the language should be checked.

We’ve performed the proof reading of the manuscript.

2. The statement - novelty of the work should already be underlined to be more attractive for the readers: in the abstract, the whole manuscript, and the conclusions.

We’ve added additional information into the Introduction to validate the novelty of the work.

3. All used chemicals to the synthesis should be described more precisely: purity, producer etc.

4. All equipment used for the measurements should be described more precisely: name, producer, country of the producer.

We’ve corrected the experimental part.

5. All presented figures are of very low quality.

Some pictures have been corrected in accordance with the recommendations of Reviewers.

6. Introduction: in this section is not well-discussed about the utilization of different forms of high-voltage materials used as electrode in Li-ion batteries. I suggest to rewrite it, and compare the results with other studies, such as: B.-Y. Lee et al. Ceram. Int. 46 (2020) 20856-20864, G. Liang et al. J. Mater. Chem. A 8 (2020) 15373-15398

Our work is mainly focused on electrolytes that can be used in high voltage batteries. The cathode based on LiNi0.5Mn1.5O4 was chosen as one of the most studied high voltage cathode materials for studying the characteristics of electrolytes. The articles mentioned by the Reviewer are completely devoted to LiNi0.5Mn1.5O4 (namely, synthesis, structural features, etc.) and they are not directly related to the topic of our article.

7. Please describe more precisely the EIS results.

We’ve added extended description.

8. Provide active material loading of LiNi0.5Mn1.5O4 (in mg/cm2) in the electrode prepared.

We’ve added this value.

9. What was the conditions used for the EIS tests, only room temperature? If yes, I suggest to perform the whole EIS analysis in high- and low-temperature conditions and re-analyze the electrolyte and additives to select the best one which could be a potential for Li-ion batteries applications.

We are going to conduct comprehensive low- and high-temperature studies of the selected electrolytes, as well as the solutions with a modified solvent composition, and hope to publish these results in the near future.

10. High-rate tests (at 5 C, 10 C, 20 C etc.) are crucial to complete the idea of this work.

We’ve added these results.

Reviewer 3 Report

In this work, the authors experimentally screened electrolytes that had been reported in different works as good choices for high voltage lithium ion batteries. However, the author did not make a more accurate conclusion and judgment on the results, and no more electrode materials were tried to verify the accuracy of the results.  Based on the current conclusions, the electrolyte screened is only used for LiNi0.5Mn1.5O4-based composite materials, and it is unknown whether it is suitable for cobalt-based cathode materials. So,  I do not recommend publication in molecules.

Author Response

We thank the Reviewer for the analysis of our manuscript. Regarding high-voltage cathode materials, in earlier works we also considered the cobalt-based compounds (see, for example, https://doi.org/10.1016/j.electacta.2018.01.037). However, current trends, which consist in the complete exclusion of cobalt from cathode materials, forces us to focus primarily on cobalt-free electrodes. The most obvious choice here is LiNi0.5Mn1.5O4, and most studies of high-voltage lithium-ion systems published in the literature are based on this material.

Reviewer 4 Report

This is an interesting work on high-voltage liquid electrolytes for lithium battery applications. The Manuscript is well written and deserve to be published in the Molecules journal after a Major Revision.

The following points must be taken into consideration during the revision process:

  1. Introduction: the authors should add a short paragraph describing polymer and ceramic electrolytes. Indeed, the main objective of this work is to discuss about the widening of the ESW, and these classes of materials would be also a good solution.
  2. Figure S1: a raise in the current value is observed already at 3.5 V for the 3M LiPF6 in EC:DEC:DMC solution. The same is observed at 4.5 V for the 2M solution. I would not recommend the use of these electrolytes in high voltage applications. Nevertheless, the authors claim in lines 93-94 that high-concentrated electrolytes are a good choice. The authors must comment on this point.
  3. Figures S2-S5: in several graphs, clear oxidation and/or reduction peaks are observed. The authors must assign these electrochemical phenomena based on the electrolyte chemical composition.
  4. Table 1: data summarized in Table 1 should be used to prepare some graphs to be added in the Supporting Information. For example, specific current values (y-axis) should be plotted vs. the lithium salt concentration (x-axis) for all the investigated salts (using different colors for each curve). Another graph could be prepares plotting the specific current values (y-axis) vs. the type of additive (x-axis). This would be of a great utility in order to better visualize the discussed trends.
  5. Lines 93-94: this is correct. Nevertheless, the opposite trend is observed at low potential. It is hard to sustain this claim, since a non-linear behavior is observed. The authors must discuss on this point.
  6. Figure 1 and Figure S1-S5: current values must be plotted as specific current (i.e., mA/g).
  7. Line 117: the accuracy of EDX analysis is known to be on the order of ca. 5 %. I suggest to reduce the number of significant figures. Otherwise, the authors should perform more accurate analyses, such as inductively coupled plasma or atomic absorption spectroscopies.
  8. Figure 3b: the Coulombic efficiency is conventionally plotted as percentage.
  9. Figure 5: Is it possible to see the whole range of analysis, maybe in the Supporting Information?
  10. Lines 189-197: The authors do not describe how the crystallography analysis was performed. What is the experimental evidence for the Mn/Ni ratio? Just the unit volume change? Could this change be addressed to any other phenomena? Literature references must be cited for the discussion of these data.
  11. The ionic conductivity of all the prepared electrolyte solutions must be determined and summarized in a Table in the Supporting Information. These values could add some important information on the different electrochemical performance.
  12. Lines 285-286: the two-electrodes cell is known to suffer from possible potential shifts. A three-electrodes cell should be used in order to avoid this kind of issues. Consider that the main objective of this work is to give a clear description of the electrochemical properties of different electrolyte solutions, and the most accurate configuration must be used for CV studies.
  13. Line 287: typo, 50 mkV s-1 instead of 50 mV s-1.
  14. The SEM images of cycled electrodes must be collected and added into the Supporting Information.

Author Response

We thank the Reviewer for an extremely attentive and scrupulous reading of our manuscript as well as for the suggestions and comments made.

  1. Introduction: the authors should add a short paragraph describing polymer and ceramic electrolytes. Indeed, the main objective of this work is to discuss about the widening of the ESW, and these classes of materials would be also a good solution.

We’ve added a brief comparison into the Discussion section.

  1. Figure S1: a raise in the current value is observed already at 3.5 V for the 3M LiPF6 in EC:DEC:DMC solution. The same is observed at 4.5 V for the 2M solution. I would not recommend the use of these electrolytes in high voltage applications. Nevertheless, the authors claim in lines 93-94 that high-concentrated electrolytes are a good choice. The authors must comment on this point.

This raise is quite low (only several µA) and most probably associated with the formation of the SEI at the first cycle. We analyzed current at the potentials above 5V, and from this point of view,the concentrated electrolytes have obvious benefits.

  1. Figures S2-S5: in several graphs, clear oxidation and/or reduction peaks are observed. The authors must assign these electrochemical phenomena based on the electrolyte chemical composition.

We fully agree with the Reviewer in an effort to understand all processes which are observing on the CV. Unfortunately, this would require a complex set of studies, such as operando Raman spectroscopy or XPS for each electrolyte solution. Such experiments are beyond the scope of this publication.

  1. Table 1: data summarized in Table 1 should be used to prepare some graphs to be added in the Supporting Information. For example, specific current values (y-axis) should be plotted vs. the lithium salt concentration (x-axis) for all the investigated salts (using different colors for each curve). Another graph could be prepares plotting the specific current values (y-axis) vs. the type of additive (x-axis). This would be of a great utility in order to better visualize the discussed trends.

We spent quite a lot of time trying to present this information graphically, but due to the abundance of curves, we abandoned this idea, since the graphs were overloaded with information and could not improve the systematization of the results. We thank the Reviewer for the idea, but the tabular representation given in the current version of the manuscript is the most convenient for the reproduction.

  1. Lines 93-94: this is correct. Nevertheless, the opposite trend is observed at low potential. It is hard to sustain this claim, since a non-linear behavior is observed. The authors must discuss on this point.

We’ve added a brief discussion of this issue.

  1. Figure 1 and Figure S1-S5: current values must be plotted as specific current (i.e., mA/g).

We’ve corrected these Figures.

  1. Line 117: the accuracy of EDX analysis is known to be on the order of ca. 5 %. I suggest to reduce the number of significant figures. Otherwise, the authors should perform more accurate analyses, such as inductively coupled plasma or atomic absorption spectroscopies.

We’ve corrected the presented data. We assume that in spite of the lower accuracy of EDX analysis compare with ICP and AAS, in our case the EDX data collected mainly form the surface and near surface electrode area provide more relevant information on Mn/Ni ratio trend caused by electrochemical processes.

  1. Figure 3b: the Coulombic efficiency is conventionally plotted as percentage.

We’ve corrected the corresponding Figures.

  1. Figure 5: Is it possible to see the whole range of analysis, maybe in the Supporting Information?

We’ve added these data to the Supplementary Materials.

  1. Lines 189-197: The authors do not describe how the crystallography analysis was performed. What is the experimental evidence for the Mn/Ni ratio? Just the unit volume change? Could this change be addressed to any other phenomena? Literature references must be cited for the discussion of these data.

Mn/Ni ratio was analyzed by EDX only. Unfortunately, the quality of the ex situ powder diffraction data obtained for the cycled electrodes using a laboratory diffractometer does not allow precise structure studies (especially with respect to Mn/Ni site occupation). We observe a correlation between cell volume change and the Mn/Ni ratio obtained by EDX. This makes it possible to assume to propose the relationship between these phenomena, but we did not find any detailed information about the change in the unit cell parameters of the LiNi0.5Mn1.5O4 cathode material after cycling.

  1. The ionic conductivity of all the prepared electrolyte solutions must be determined and summarized in a Table in the Supporting Information. These values could add some important information on the different electrochemical performance.

We did not determine the ionic conductivity of the electrolytes since it is not directly related to their high voltage stability. Unfortunately, at present there are certain technical difficulties in performing this work, associated with the degradation of some electrolytes during storage. All we can do now is take into account the Reviewer's remark for the future.

  1. Lines 285-286: the two-electrodes cell is known to suffer from possible potential shifts. A three-electrodes cell should be used in order to avoid this kind of issues. Consider that the main objective of this work is to give a clear description of the electrochemical properties of different electrolyte solutions, and the most accurate configuration must be used for CV studies.

We have analyzed the electrochemical responses in two- and three-electrode cells and found that the difference is negligible for the certain types of experiments (for example, galvanostatic charge-discharge). We apply a three-electrode configuration in the cases where voltammetric curves are used to obtain numerical characteristics of the processes (for example, diffusion coefficients or contributions from diffusion or surface processes, https://doi.org/10.1016/j.electacta.2020.136647).

  1. Line 287: typo, 50 mkV s-1 instead of 50 mV s-1.

We’ve corrected the text.

  1. The SEM images of cycled electrodes must be collected and added into the Supporting Information.

We’ve added these images to Figure 5.

Reviewer 5 Report

The present Paper shows a study of the electrochemical stability of various electrolyte solutions for LIBs, showcasing trends as a function of the concentration, dissolved species and they analyze the impact of them by cycling against a TMO cathode. The study is globally good, data are provided with clearlity. However, some minor comments that should be considered by the Authors:

Figure 2 c the Authors can change the color/shape of either charge or discharge profile, to help the reader (not familiar with irreversible capacity) to easily recognize them in a b/w print version of the paper

Page 6 line 163 should be SEI and not CEI (page 9 line 217 too). Moreover in that sentence, is it rather unclear and not well supported by opportune references, the generic statement of a "different mechanism of SEI formation... ... due to the occurrence of competing processes with distinct rate constant" It would be helpful for the Reader to bo aware of those processes are.

Author Response

We thank the Reviewer for the analysis of our manuscript

Figure 2 c the Authors can change the color/shape of either charge or discharge profile, to help the reader (not familiar with irreversible capacity) to easily recognize them in a b/w print version of the paper

We’ve corrected the Figure.

Page 6 line 163 should be SEI and not CEI (page 9 line 217 too). Moreover in that sentence, is it rather unclear and not well supported by opportune references, the generic statement of a "different mechanism of SEI formation... ... due to the occurrence of competing processes with distinct rate constant" It would be helpful for the Reader to be aware of those processes are.

We’ve corrected the abbreviation and added some explanation regarding cathodic SEI.

Round 2

Reviewer 3 Report

This version can be considered for acceptance.

Author Response

We thank the Reviewer for the analysis of our revised manuscript and for the appreciation of our work.

Reviewer 4 Report

The authors did not provide a sufficient revision of the manuscript. The work did not reach a sufficient scientific level for publication. Thus, I suggest to reject this work.

My main concerns are summarized here below:

  1. The authors did not provide a suitable answer for question # 3, 4, 9, 11, 12.
  2. I would like to point out that answer to question #9 is very disappointing, since the authors were asked to add the XRD curves in the whole range (Figure) and, differently from what they answered, no additional figures were added to the Supplementary Information. At least, I do not see any XRD curve in the Supplemetary Information.
  3. The novel text added in order to answer to question #1 does not correctly describe the majority of the polymer and ceramic electrolytes. The authors just provide a description of a limited number of materials exhibiting a low performance, maybe to emphasize the properties of the proposed materials.
  4. In answer #2 the authors minimize the detected current values, addressing them to the SEI formation. Nevertheless, a quite low Coulombic efficiency is observed in Figure 3b, which clearly indicates that the current values detected in CV experiments should not be attributed just to the SEI formation, but to clear oxidative degradation events.
  5. In answer to question #11 the authors state “Unfortunately, at present there are certain technical difficulties in performing this work, associated with the degradation of some electrolytes during storage”. This means that they were aware of the impossibility to perform any additional experiment requested from reviewer. From the scientific point of view, the instability of the materials should be clearly stated in the manuscript, since it is a fundamental property of the electrolytes.
  6. “All we can do now is take into account the Reviewer's remark for the future”: I am asking precise questions in order to improve the quality of THIS manuscript, not the future works of the authors.
  7. Answer to question #12 is not consistent to what is declared in the Experimental Section. Which one is the correct one?

Author Response

The authors did not provide a sufficient revision of the manuscript. The work did not reach a sufficient scientific level for publication. Thus, I suggest to reject this work.

We thank the Reviewer for the analysis of our revised manuscript

My main concerns are summarized here below:

The authors did not provide a suitable answer for question # 3, 4, 9, 11, 12.

I would like to point out that answer to question #9 is very disappointing, since the authors were asked to add the XRD curves in the whole range (Figure) and, differently from what they answered, no additional figures were added to the Supplementary Information. At least, I do not see any XRD curve in the Supplemetary Information.

We apologize for this technical error. These data are added into the updated version of the SI file.

The novel text added in order to answer to question #1 does not correctly describe the majority of the polymer and ceramic electrolytes. The authors just provide a description of a limited number of materials exhibiting a low performance, maybe to emphasize the properties of the proposed materials.

Since our study and competence area are devoted to liquid electrolytes only, we do not feel entitled to present a serious review of polymer and ceramic electrolytes. Moreover, for this work, it seems to us redundant. There are many articles in the literature, including reviews, written by experts in this field.

In answer #2 the authors minimize the detected current values, addressing them to the SEI formation. Nevertheless, a quite low Coulombic efficiency is observed in Figure 3b, which clearly indicates that the current values detected in CV experiments should not be attributed just to the SEI formation, but to clear oxidative degradation events.

The reviewer spoke in his question 2 about the currents that occur at a potential of 3.5 V in 3M LiPF6 solution. As one can see from the charge-discharge curve below (in the attachement), the cell does not gain any significant charge at this potential (the sloping curve between 3.75-4.5 V is a characteristic feature of the Fd-3m-type LiNi0.5Mn1.5O4 and does not refer to electrolyte degradation).

In answer to question #11 the authors state “Unfortunately, at present there are certain technical difficulties in performing this work, associated with the degradation of some electrolytes during storage”. This means that they were aware of the impossibility to perform any additional experiment requested from reviewer. From the scientific point of view, the instability of the materials should be clearly stated in the manuscript, since it is a fundamental property of the electrolytes.

Instability of the solutions is quite a complex issue that is extremely rare discussed in the academic research papers. Researchers working experimentally in the field know that it is best to work with freshly prepared electrolytes, especially if it contains some little-studied additives. We have noticed that some solutions look different after several months spent in the glovebox, and exhibit different electrochemical characteristics, but some are quite stable. However, a detailed study of the degree of degradation of various solutions depending on the conditions of their storage is part of a technological, not a scientific process, and is clearly beyond the scope of a scientific publication

“All we can do now is take into account the Reviewer's remark for the future”: I am asking precise questions in order to improve the quality of THIS manuscript, not the future works of the authors.

We thank the Reviewer for the intention to improve our manuscript

Answer to question #12 is not consistent to what is declared in the Experimental Section. Which one is the correct one?

In our opinion, the answer is quite consistent with what is written in the experimental part. As we mentioned in the answer, we apply a three-electrode configuration in the cases where voltamperic signal is used to obtain numerical characteristics of the processes such as diffusion coefficients or contributions from diffusion or surface processes based on the analysis of current values at different potential sweep rates. In the present work, we did not aim at a deep analysis of the current-voltage response at various cycling rates; therefore, there was no need for a three-electrode cell configuration. All the CV experiments were performed at a single and quite slow rate of 50 µV/s, and the contribution of the Li electrode resistance is negligible.

Round 3

Reviewer 4 Report

Satisfying answers were provided only to questions #2 (XRD) and #7 (inconsistency in the Experimental section: during R1 I understood that some of the experiments done in this work were performed with a 3-electrodes cell, and others with a 2-electrodes cell. Now it is clearer this point). All the other responses are not enough to me, and my suggestion is still to reject this work.

Let me say also that I fully agree with the authors that the degradation of the materials has more a technological aspect, rather than a scientific one. Moreover, I appreciate (and this is how I also work) that authors study only freshly prepared solutions in order to provide an exact description of the materials. Nevertheless, my opinion is that the novel studies requested by the Reviewer fall within the scientific process of materials studying, and not the technological. Conventionally, in these cases, the materials should be freshly re-synthetized, checked to confirm their reproducibility, and then the requested novel studies can be performed and added during the revision process. At least, this is my opinion.